# Optimizing testing for COVID-19 in India

**Philip Cherian**[1], **Sandeep Krishna**[2]*, **Gautam I. Menon**[1,3,4,5]*

**1** Department of Physics, Ashoka University, Sonepat, Haryana, India, **2** Simons Centre for the Study of Living Machines, National Centre for Biological Sciences, Tata Institute of Fundamental Research, Bangalore, Karnataka, India, **3** Department of Biology, Ashoka University, Sonepat, Haryana, India, **4** Theoretical Physics and Computational Biology Groups, The Institute of Mathematical Sciences, Taramani, Chennai, India, **5** Homi Bhabha National Institute, Mumbai, India

* sandeep@ncbs.res.in (SK); gautam.menon@ashoka.edu.in (GM)

**Data Availability Statement:** All relevant data are within the manuscript and its Supporting information files. The current source code of the model used in the simulations can be found here: https://github.com/dpcherian/optimizing-testing-COVID-19.

## Abstract

COVID-19 testing across India uses a mix of two types of tests. Rapid Antigen Tests (RATs) are relatively inexpensive point-of-care lateral-flow-assay tests, but they are also less sensitive. The reverse-transcriptase polymerase-chain-reaction (RT-PCR) test has close to 100% sensitivity and specificity in a laboratory setting, but delays in returning results, as well as increased costs relative to RATs, may vitiate this advantage. India-wide, about 49% of COVID-19 tests are RATs, but some Indian states, including the large states of Uttar Pradesh (pop. 227.9 million) and Bihar (pop. 121.3 million) use a much higher proportion of such tests. Here we show, using simulations based on epidemiological network models, that the judicious use of RATs can yield epidemiological outcomes comparable to those obtained through RT-PCR-based testing and isolation of positives, provided a few conditions are met. These are (a) that RAT test sensitivity is not too low, (b) that a reasonably large fraction of the population, of order 0.5% per day, can be tested, (c) that those testing positive are isolated for a sufficient duration, and that (d) testing is accompanied by other non-pharmaceutical interventions for increased effectiveness. We assess optimal testing regimes, taking into account test sensitivity and specificity, background seroprevalence and current test pricing. We find, surprisingly, that even 100% RAT test regimes should be acceptable, from both an epidemiological as well as a economic standpoint, provided the conditions outlined above are met.

## Author summary

Using network models, we study optimal ways of combining low sensitivity, relatively inexpensive point-of-care rapid antigen tests for COVID-19 with higher sensitivity but more expensive laboratory RT-PCR tests. We take into account background seroprevalence and current test pricing for such tests in India, finding that even purely rapid antigen test-based regimes can produce the same reduction in overall infections that pure RT-PCR tests are capable of. This is provided one can test at scale and isolate those testing positive effectively, that the sensitivity of the rapid test is not too low and that non-pharmaceutical interventions proceed in parallel for increased effectiveness.

**Funding:** GIM acknowledges support of the Bill and Melinda Gates Foundation, Grant No: R/BMG/PHY/GMN/20, while PC acknowledges support from Ashoka University. SK acknowledges support of the Department of Atomic Energy, Government of India, under Project Identification No. RTI 4006, and of the Simons Foundation. The funders had no role in study design, data collection and analysis, decision to publish, or preparation of the manuscript.

## Introduction

The number of COVID-19 cases in India increased at a relatively slow rate in the first few weeks after the first case was recorded on January 30, 2020. A stringent, country-wide lockdown, imposed on March 24, stretched to nearly 70 days [1]. After the lockdown was lifted, the pace at which new cases were added accelerated, with daily new cases reaching a peak of just under 98,000 by the middle of September. There was a subsequent sustained decline in incidence until the middle of February 2021 when a second wave began, with a far steeper onset than the first. New and more transmissible variants are believed to be driving this second wave [2].

Multiple serological studies inform our understanding of the first wave of COVID-19 spread in India, among them two large-scale serosurveys conducted by the Indian Council of Medical Research (ICMR) [3]. These indicate that about 0.73% (95% *CI*: 0.34—1.13) of the adult Indian population may have been infected by early May, 2020. This fraction is estimated to have risen to 7.1% (95% *CI*: 6.2—8.2) by August, 2020 [3]. Such studies, alongside others, indicate that COVID-19 spread across India has been extremely inhomogeneous. Among the large Indian cities, for Mumbai, a seropositivity of 57.8% (slums) and 17.4% (non-slum) was reported in a study conducted between 29 June and 19 July, 2020 [4]. In Delhi, a seropositivity of 23% was obtained in a study conducted between 27 June and 10 July, 2020 [5]. A subsequent study in Delhi, conducted in early August, obtained a seropositivity of 29.1% [6]. For Pune, in a study conducted between July 20 and Aug 5, 2020, an overall seropositivity of 51.3% was reported [7]. For the state of Karnataka, an overall infected population fraction of 27.3%—with that fraction including a large number of active cases—was obtained in a study conducted from early to mid-September, 2020 [8]. These studies point to extensive spread, dominated by the urban agglomerations that account for about 30% of the Indian population [9].

Surprisingly, the states of Uttar Pradesh (UP) and Bihar saw relatively small numbers of cases relative to the size of their population in the first wave [10]. Whether case and mortality numbers in these states reflect the underlying reality is a topic of some current debate [11]. The importance of these states derives from their sheer size: if the state of UP were an independent nation it would be the fifth most populous in the world [12]. (Every country in Africa, Europe, and South America has fewer people than the state of UP.) Bihar, India's third most populous state, is the third largest sub-national entity in the world by population [13]. Bihar ranks at the bottom of India's state-level SDG index while UP fares only a little better, at five slots above [14, 15]. Thus, understanding mitigation strategies for COVID-19 in these states may hold broader lessons for other parts of the developing world, in particular for low and middle income countries (LMICs).

Before vaccines become generally available, testing at scale for the presence of infection can blunt the progress of the pandemic. Currently—the third week of April 2021 at the time of writing—India is reporting close to 300,000 cases, with test positivities reported to be in excess of 15% nationwide. Questions surrounding the best way to test have assumed added urgency.

Testing, in general, serves two purposes. For clinical purposes, it concentrates on identifying disease in symptomatic patients, while testing for epidemiological purposes attempts to identify disease in patients who may be asymptomatic as well, so that spread in the population can also be assessed. India was testing 1—1.5 million people each day, less than 0.1% of the Indian population of approximately 1.34 billion, at the peak of the first pandemic wave. About 49% of these tests were RAT tests, principally the SD Biosensor test (SD Biosensor Standard Q COVID 19 Antigen test). This was the first to be certified by ICMR [16–18]. The remaining 51% were very largely RT-PCR tests. (TrueNat and CBNAAT tests, both PCR-based tests that use a technology developed for tuberculosis testing, provided a small fraction, of less than 5%, towards the total.).

While test availability was a major initial concern, the issue of sensitivity of these tests has attracted more recent attention. RATs have lower analytical sensitivity than PCR-based tests, but a comparable specificity. For an RAT kit to be approved in India, the ICMR requires it to have a minimum specificity of 95% and a sensitivity of 50% [16]. An initial advisory from the ICMR indicated that the SD Biosensor RAT performance, as measured in two independent labs, showed a sensitivity of 50.6% and 84%, with a specificity of 99.3% and 100%, respectively [16]. A more recent field evaluation yielded a sensitivity and specificity of 70.0% (95% CI: 60—79) and 92% (95% CI: 87—96) respectively [19]. An additional study of the same test obtained 76.6% (95% CI 63–86) sensitivity and 99.3% (CI 98.6—99.6) specificity [20].

It has been suggested that low case numbers in the first wave of COVID-19 in the states of UP and Bihar reflected that fact that the rapid antigen tests were missing infections [18, 21, 22]. RAT positivity in both UP and Bihar was less than 1%, while the RT-PCR positivity was around 2.8% [23]. This difference attracted concern, since in both these states the fraction of RATs *vs*. RT-PCR tests was larger than the fraction across India, with RAT fractions of approximately 59% (UP) and 87% (Bihar) [21, 24]. There were calls for the reduction in use of RATs and for their replacement by PCR-based tests. Sustained pressure from the Indian states and central government led to steep reductions in the costs of the RT-PCR tests, although they remain more expensive than the RATs. A broader concern with the RT-PCR tests are delays in reporting. These delays have been reported to be anywhere between 24 hours to a week or more [25]. In the second wave of the pandemic, such delays have only exacerbated.

Testing regimes that are accurate, comprehensive and affordable are clearly preferable to those that fail on one or more of these counts. However, what is possible in practice is often only some reasonable triangulation between these. Understanding what testing regimes are optimal for India requires reasonable estimates of how much testing levels can be ramped up, the relative costs of the two types of tests, the mix of the two tests, as well as a target for the reduction of new infections through such epidemiological screening.

This paper explores, through computational models, the epidemiological consequences of different testing regimes that combine RAT and RT-PCR tests, in a situation where population seroprevalence is of order 20%. (This is a reasonable estimate for the Indian states of UP and Bihar for much of the latter part of the first wave. It may even be relevant to the second wave if reinfections are significant, although this is unclear currently.) We use a network model that combines a compartmental model of infection spread with movement of individuals between different locations. Our testing protocol prioritizes symptomatic individuals for testing. It uses any tests left over on the remaining population at random till all tests allocated for that day are exhausted. We vary test rates as a fraction of the population between 0.1% and 1.0% per day, the test sensitivity across values believed to be accurate for the RAT tests, and the mix of RAT and PCR tests given a total amount of testing, ranging between all RT-PCR to all RAT tests. (RT-PCR tests are treated as the gold standard, with 100% sensitivity and specificity.) We quantify tradeoffs between different testing regimes by evaluating the total number of people infected at the end of the pandemic under each regime, comparing this to the scenario where no interventions are made. This helps us calculate optimal testing scenarios required to reach a specified suppression of the pandemic, for example a reduction in the total numbers of infected from 65% to 50%.

## Model and methods

### Network model description

We consider a population of individuals with each individual classified as belonging to one of 7 possible states. These states are: (S)usceptible, (A)symptomatic, (P)resymptomatic, Mildly

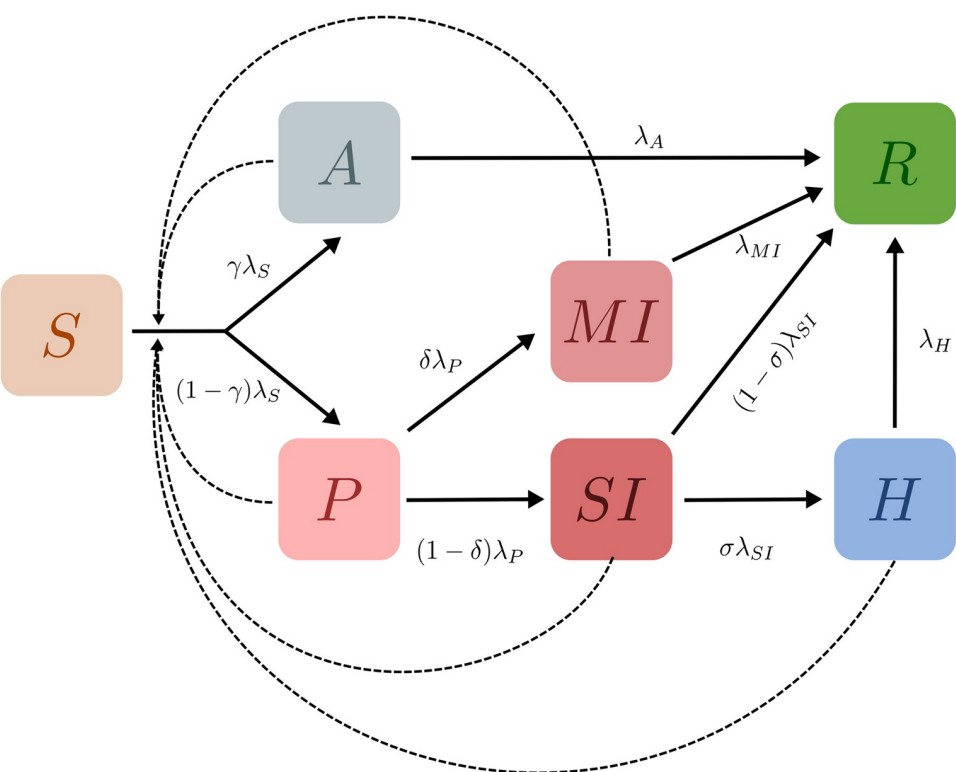

**Fig 1. Schematic of the epidemiological model.** Boxes show the 7 compartments of the model, i.e., the set of states that characterise each individual in the population. The compartments are (S)usceptible, (A)symptomatic, (P) resymptomatic, Mildly Infected (MI), Severely Infected (SI), (H)ospitalized, and (R)ecovered. Black arrows indicate possible transitions between states. The dotted lines indicate that the rate of infection of Susceptible individuals is an increasing function of the number of infected individuals they are in contact with. Recovered individuals are assumed to be immune to further infection.

Infected (MI), Severely Infected (SI), (H)ospitalized, and (R)ecovered. These are shown in Fig 1. Black solid lines with arrows indicate transitions between states, while dotted lines denote influences. We assume in this paper that recovered individuals are immune to further infection. We ignore deaths, although they could be easily incorporated into this general framework [26].

If all individuals were in a single well-mixed location, this would correspond to a 7-compartment epidemiological model that generalizes standard SIR/SEIR-type models to incorporate additional distinctions, such as those between asymptomatic and symptomatic individuals (see S1 Appendix for an analysis of this compartmental model). Here, we incorporate these elements into a network model [27–29], wherein individuals exist in, and move between, multiple locations. Our schematic network is shown in Fig 2. It shows two categories of locations: homes and workplaces. A subset of workplaces are designated as hospitals.

Fig 3 shows how disease dynamics proceed on the network i.e. how the aggregate numbers of susceptible, infected and recovered individuals changes with time. The state of the network is shown at three different times, from early to late in the epidemic. In each graph, the nodes represent locations: nodes indicated in grey are homes and black nodes are work locations. Hospitals are shown as blue nodes. Undirected links represent individuals and connect the home of that individual to their work location. The color of these links represents the infection

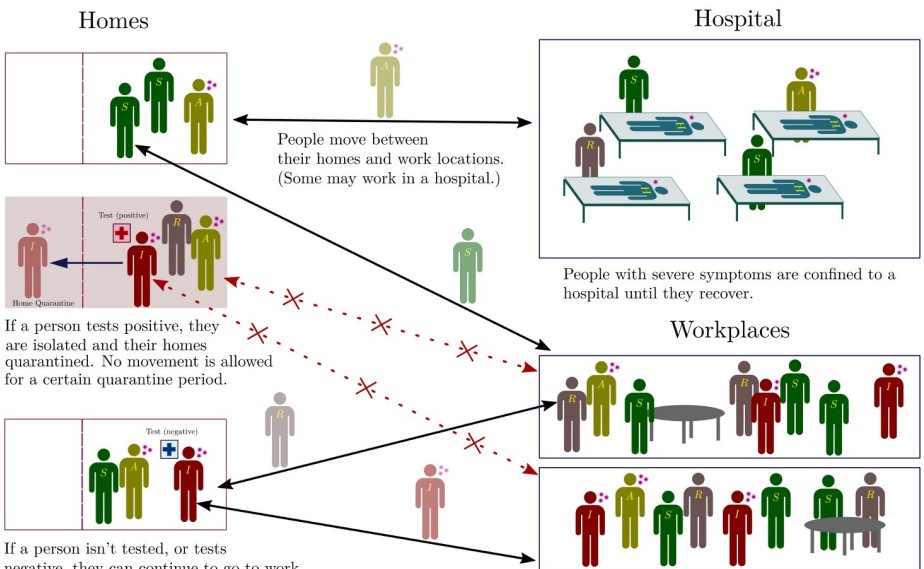

**Fig 2. Schematic of the network.** Individuals exist on a location network, consisting of homes and work places, connected by links along which people can move. For each individual there is a link between their home and their work place. Within each location, individuals can transition between the different infection states described in Fig 1. A subset of work places are designated as hospitals, where people are moved (and quarantined) when they transition from the *SI* to the *H* state. When they recover, they are moved back to their homes and can resume movement between their home and work. In addition, if a person tests positive they and their home are quarantined, with no movement being allowed out of the home for 14 days (we explore several variants of this protocol as described in subsequent sections).

state of the individual. Yellow links represent susceptible individuals, red links represent those infected and green links represent those who have recovered.

## Network model implementation

Each individual has one home and one workplace. (We make this assumption for simplicity, but more complex networks, where individuals move between several locations, can just as easily be modelled within our framework.) Within a 24-hour cycle, all individuals move once from home to workplace and back. We consider individuals whose workplace happens to be a hospital as designated health-care workers. Each location is considered to be a well-mixed system.

At any given time, each location represents a fully connected network of individuals. However, the number of individuals in any given location varies as they move from homes to workplaces and back. In the absence of testing, individuals transition stochastically, at specified rates, from their current state to other accessible states. These rates are illustrated in the progression shown in Fig 1. The values we use are specified in Table 1. Except where explicitly mentioned, we use the indicated default values for transition rates and probabilities for branching between possible states [30–36].

Note that the rate of exit from each state, except *S*, is a constant. The force of infection, that is the rate at which each individual susceptible person becomes infected, is however proportional to the product of the number of infectious people at their current location and the infectivity parameter $\lambda_S$, divided by a normalization factor which is related to the local population at that location as described below. This corresponds to frequency-dependent scaling of transmission [41–43]. Testing, quarantining, and hospitalization of individuals add further layers of

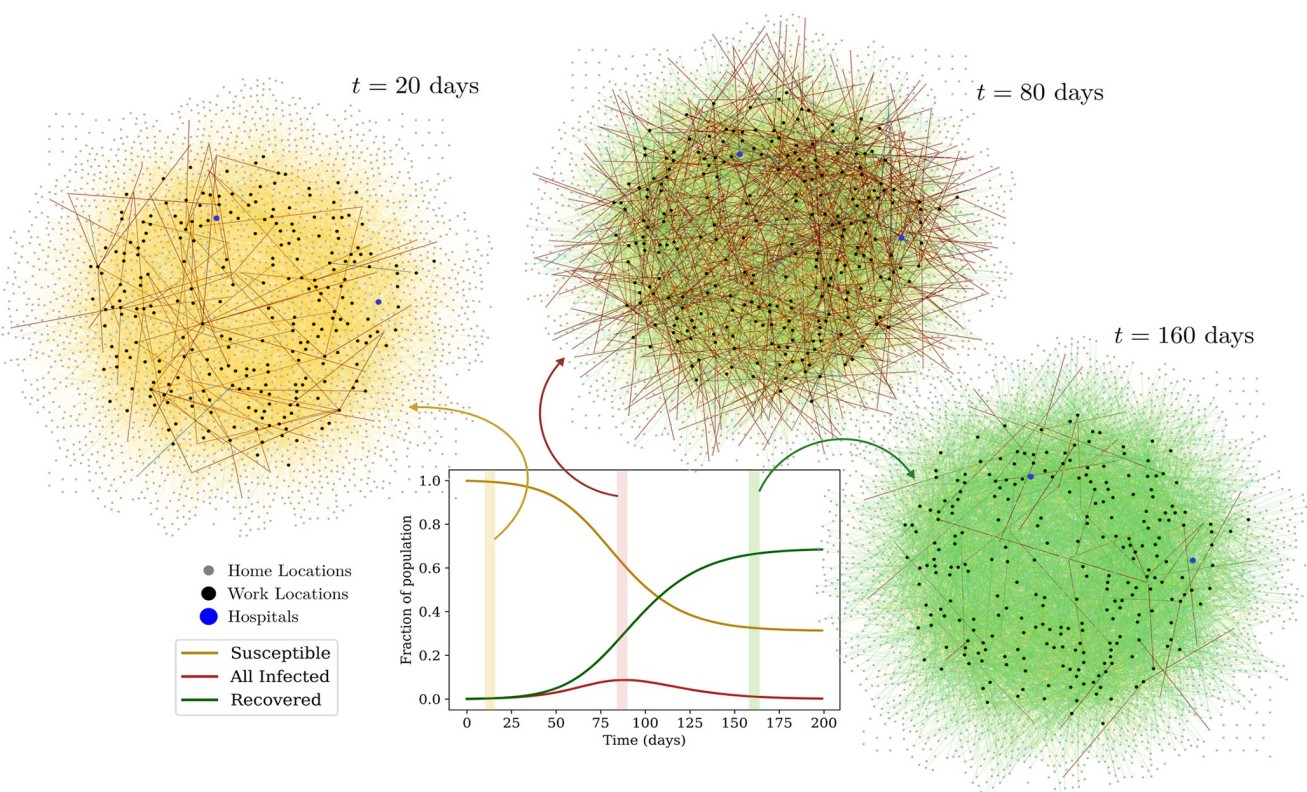

**Fig 3. Visualization of the network.** The plot shows the aggregate number of susceptible, infected ($A + P + SI + MI + H$) and recovered individuals across all locations as a function of time. In the three graphs, the nodes represent locations; the grey nodes are homes and the black nodes are work locations. Hospitals are represented by blue nodes. Individuals are represented by (undirected) links between a home and a work location. The color of these links represents their infection state: Yellow links represent susceptible individuals, while red links represent the infected across the infected compartments, and green links represent the recovered.

complexity to the model, as described in more detail in subsequent sections. If an individual transits into the Hospitalized state they are immediately moved to a randomly chosen hospital, where they remain until they have recovered and are moved back to their home location.

Individuals who test positive are confined to their homes for 14 days (chosen to be longer than their average recovery time). In this case, they are no longer allowed to move to their

**Table 1. The parameters used to specify the model include both rates for transitions between compartments as well as the probabilities of branching between them.** These are within ranges found in the literature. We ignore deaths, since reported IFRs for COVID-19 in India are small ($< 0.1\%$). These are the parameters used for the simulation depicted in Fig 3.

| Parameter | Value | Description | Source |
|---|---|---|---|
| $\lambda_S$ | 0.25 | Rate at which infected individuals can infect the susceptible population | Ref. [32] |
| $\gamma$ | 0.5 | Fraction of infected population that remains asymptomatic until recovering | Ref. [33, 34] |
| $\lambda_A$ | 0.143 | Asymptomatic individuals remain infected for an average of $1/\lambda_A$ days | Ref. [33, 34] |
| $\lambda_P$ | 0.5 | Presymptomatic individuals remain infected for an average of $1/\lambda_P$ days | Ref. [37–39] |
| $\delta$ | 0.85 | Fraction of presymptomatic individuals who transition to being mildly infected | Ref. [40] |
| $\lambda_{MI}$ | 0.1 | Individuals experiencing mild symptoms recover in an average of $1/\lambda_{MI}$ days | Ref. [35, 36] |
| $\lambda_{SI}$ | 0.5 | Individuals experiencing severe symptoms recover in an average of $1/\lambda_{SI}$ days | Ref. [35, 36] |
| $\sigma$ | 0.8 | Fraction of severely infected individuals who are hospitalised | Ref. [36] |
| $\lambda_H$ | 0.1 | Individuals who are hospitalized recover in $1/\lambda_H$ days on average | Ref. [36] |

work locations for the duration of confinement. Furthermore, their overall infectivity is reduced by a factor of ten, to simulate limited contact with the others in their family. The same holds true for hospitalized individuals, since we assume that measures have been put in place in hospital locations to ensure that individuals do not leave until they have recovered, and are less infectious.

The force of infection at any location is given by [41]:

$$\text{Force of infection at location } i = \frac{\lambda_S}{V_i}\left(N_{\text{inf}} + C_H H_i + C_Q N_{\text{inf}}^{\text{conf}}\right),$$

where $H_i$ the number of hospitalized currently in location $i$, and $N_{\text{inf}}$ and $N_{\text{inf}}^{\text{conf}}$ denote the number of infected individuals (of all states except hospitalized) in location $i$ that are not confined and are confined, respectively. This expression is used to compute the rate at which each susceptible individual in any given location will transition to the asymptomatic or pre-symptomatic state.

The contact parameters $C_H$ and $C_Q$ that scale the infectivity are both chosen to be 0.1 (varying these numbers does not produce qualitatively different results, but it does change the overall fraction of healthcare workers (HCWs) who become infected as compared to non-HCWs. We make this choice so that, in the absence of testing, HCWs are up to 25—30% more likely to become infected compared to non-HCWs; related numbers are reported in [44, 45].) The quantity $V_i$ is chosen to be the total number of individuals currently in that location $N_i$ in all locations except for hospitals, where this number is changed to an effective $N$, related to the total number of healthcare workers as follows:

$$V_i = \begin{cases} N_{\text{HCW}} + C_H H_i & \text{if } i \in \text{ Hospitals}, \\ N_i & \text{otherwise}. \end{cases}$$

For example, a hospital location with 40 healthcare workers and 60 hospitalized individuals will have a total normalization factor of $(40 + 0.1 \times 60) = 46$, instead of 100. Here, assigning hospitalized individuals a tenth of the weight in the normalization factor, as compared to other individuals, accounts for the fact that hospitals have measures in place to reduce infectivity, such as the compulsory use of PPE.

## Models for testing and quarantining

In the simulations whose results we discuss below, the following testing protocols are employed.

**Eligibility for testing.**  Except where mentioned otherwise, testing is started when 20% of the population has recovered from the disease. Each day a fixed number of tests are performed, determined by the testing rate parameter. A fraction of these are RATs, determined by the RAT:PCR ratio parameter. PCR tests are used first. Then, once all PCR tests for the day are exhausted, RATs are used. All tests, PCR or RAT, are targeted at symptomatic individuals across all locations (both severely and mildly infected), who are given first preference for the tests. (We found purely random testing to be highly inefficient at the daily testing rates considered, as discussed in S3 Appendix.) For an individual to be eligible for targeted testing, they would have to not be currently awaiting a result of a test nor be currently confined because of a positive result less than 14 days ago. Note that hospitalized individuals are not tested, since they are already confined and isolated in hospitals.

Among those that are being tested in this fashion, symptomatic individuals who receive a negative result in an RAT test over 7 days ago are given first preference, i.e., they will almost

certainly receive a PCR test if available. (We have also run simulations without such prioritization, and found that it made little difference at the testing rates that we examined.) Once all eligible symptomatic individuals are tested on any given day, any remaining tests are distributed randomly among the eligible individuals in the remaining population. (At the testing rates we explore, these random tests are mostly RAT and not PCR whenever there are a substantial number of symptomatic individuals who remain to be tested, because PCR tests are used first.) To be eligible for random testing an individual must (i) not currently be hospitalized, (ii) not be awaiting a result of a test and (iii) not currently be confined because of a positive result less than 14 days ago.

**Mixtures of tests.** We study mixtures of tests, i.e. combinations of overall fractions of RAT and PCR tests, as discussed above. Test sensitivities are varied as follows: The PCR test is assumed to have 100% sensitivity and specificity, while the RAT test sensitivity is varied from 50% to 95% in steps of 5%. The RAT specificity is fixed at 98%.

We can also account for test delays, quantifying this in terms of the number of days that intervene till the result is declared. We compare between mixtures that have no test delay (results declared immediately) and mixtures that have a PCR test delay of 5 days i.e. results for PCR tests are declared 5 days after the sample is taken. We will assume RAT tests to be point-of-care and hence that their results are obtained immediately.

**Isolation and quarantine strategies.** The consequences of testing can be examined using the different isolation and quarantining strategies listed below:

(a). **Isolate when result declared**: Individuals who tested positive are isolated at their homes for 14 days and not allowed to leave this location. They are assumed to be 10-fold less infectious, and therefore 10-fold less likely to infect their family members. No additional constraints are placed on the family.

(b). **Home quarantine when result declared**: Individuals who test positive are isolated at their homes. All family members are also quarantined for 14 days from the day the result is declared.

(c). **Home quarantine when sampled**: In this case both the individual and the home are quarantined when the test sample is taken, until the result is declared. If the result is positive, the quarantine is extended for a further 14 days.

Strategy (c) may be difficult to implement practically when tests are randomly distributed (as happens towards the later part of the epidemic) and the individual and others in their home may not exhibit any symptoms. It would be easier to justify when the individual sampled is symptomatic. Nevertheless, we discuss this case so that the effects of a more stringent strategy can also be studied. We have also considered the **Isolate when sampled** case in which individuals who are tested are immediately isolated (as above) until the result is declared, but family members from the same home are not confined. If the result is positive, these individuals are isolated for a further 14 days. While omitted here for the sake of compactness, we provide related results and discussion in S4 Appendix.

## Details of the numerical calculation

Individual agents are assigned a number of attributes, including their infection state, home and work locations, current location, status of confinement, last test result, date last tested, date last isolated, and so on. Each location has a similar set of attributes, such as the number of individuals in every state in that location, the quarantine status of the location, and the date it was last quarantined. We use a standard Monte Carlo algorithm where each individual transits

between compartments (infection states) with an exponentially distributed time spent in each compartment.

While the model as defined is a stochastic continuous-time model, we implement it numerically using a discrete-time approximation with a very small time step (in our case, `dt = 0.01 days`). Individuals transit out of the compartment $i$ they are currently in with a probability of $\lambda_i dt$. A second random number is chosen in the case of branching to decide into which compartment the individual transits. As a check on our results, we also experimented with Gillespie methods, including the tau-leap algorithm, but decided finally that the requirements of large population sizes, extensive averaging and the complexity of the dynamics favoured simpler Monte Carlo methods. However, we verified that results obtained with these alternative methods were both qualitatively and quantitatively consistent with our Monte Carlo results.

Our networks consist typically of 10,000 individuals and 2,750 locations, although a number of our simulations were carried out for 100,000 individuals and 27,500 locations to check for finite-size effects. (We found that these simulations produced similar qualitative trends, and even quantitatively similar proportions of total infected individuals; see S2 Appendix). In all simulations, we chose the ratio of home to work locations to be 10:1, resulting in the number of individuals per home distributed as a Poisson distribution with mean 4, and the number per work location distributed as a Poisson with mean 40.

We move individuals between home and work locations synchronously at 12 hour intervals. (Note that the rates specified in Table 1 are for a day i.e, 24 hours. Our update probabilities are scaled accordingly.) Once a target fraction of recovered has been crossed, at the end of every day the population is tested with a fixed number of tests. Then any pending results are declared, the numbers of the different tests available daily are reset, and the entire process repeats.

A flowchart of our model implementation is shown in S8 Appendix.

## Results

### Effects of starting testing later in the pandemic

We study the effect of testing strategies that are implemented once a certain fraction of the population has already recovered from the disease, as shown in Figs 4 and 5. As expected, starting testing early on in the pandemic has a greater effect in reducing the infection curves, bringing down the overall peak of incidence and shifting it earlier, as explained in the caption. The general trend is globally the same for RAT and PCR tests, with low sensitivity RAT tests being less efficient in shifting the peaks.

### Effects of RAT test sensitivity and different test mixtures

The effects of varying both the RAT test sensitivity as well as the ratio of RAT to PCR in the test mixtures are shown in Fig 6, when the test result is declared immediately (no test delay). Higher testing rates lead to a greater suppression in the total infected fraction.

However, as can be seen in Fig 7, the benefit does not scale linearly with an increase in testing rates—increasing testing rates has a greater overall effect at lower testing rates than at higher testing rates. The mixture of tests matters less as we start testing later in the pandemic. However, for any fixed value of daily testing, the test ratios do matter. We find that RAT:PCR ratios even as skewed as 80:20 give reasonable results for mixtures of tests in which the RAT tests have the lowest sensitivity in the range we consider.

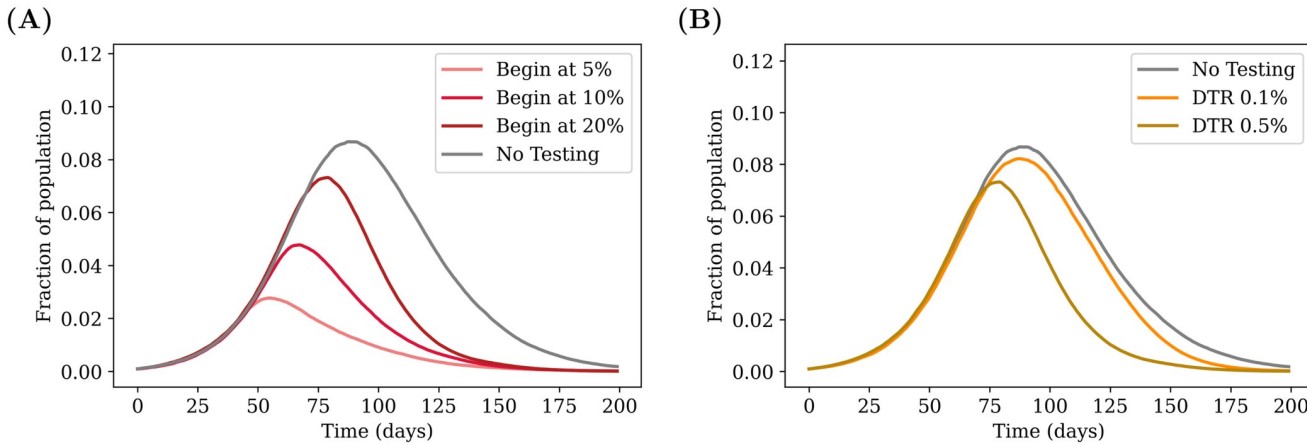

**Fig 4. Effect of targeted testing.** (A) Comparing the effect of delays in starting testing. The daily testing rate is fixed at 0.5% of the population per day, but testing is started at different points during the infection, depending on the number of recovered individuals present in the population. The greatest advantage is obtained when testing is started earlier. (B) Varying the daily testing rate. Testing is started when 20% of the population has recovered from the disease, and the effects of different testing rates are compared. Higher testing rates reduce the value of the peak and push it earlier in the infection. The grey curves in both cases represent the case of no testing. All tests are PCR (sensitivity and specificity of 100%) with results declared immediately (zero delay), and individuals (but not family members) are isolated when a test is declared positive (see strategy (a) in Methods). The curves are obtained after averaging over 100 simulation runs.

## Effects of different testing models and quarantine strategies

In this subsection, we compare different testing strategies across different daily testing rates and test delays, starting once 20% of the population has recovered. We conclude the following: If there is no delay in receiving test results, the largest reduction in the total number of infected people (equivalent to the final number of recovered) is obtained by increasing the daily

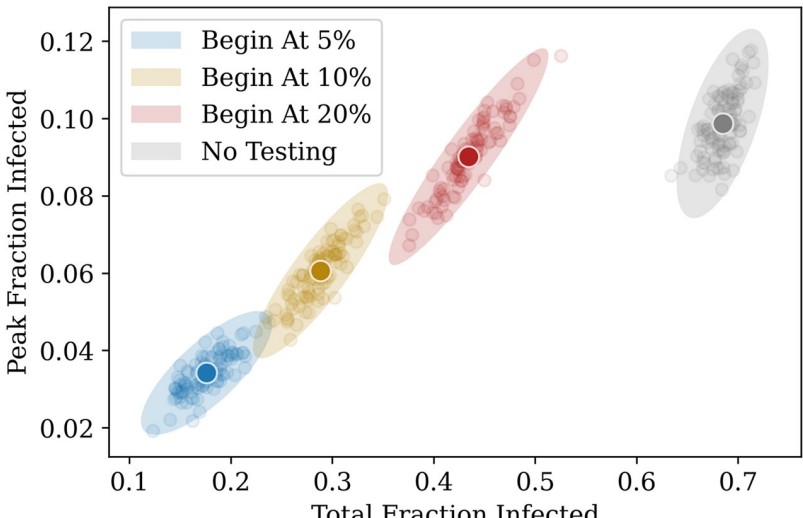

**Fig 5. Another representation of Fig 4A.** The daily test rate is fixed at 0.5% of the population per day, and testing is started at different fractions. The *x*–axis represents the total fraction who have contracted the illness by the end of the pandemic, and the *y*–axis represents the peak infected fraction. Each point represents the result of a single simulation. A total of 100 simulation runs of 200 days each were performed for each case. The ellipses are centered around the average point (highlighted) and their radii represent 3 standard deviations of the distribution across simulations.

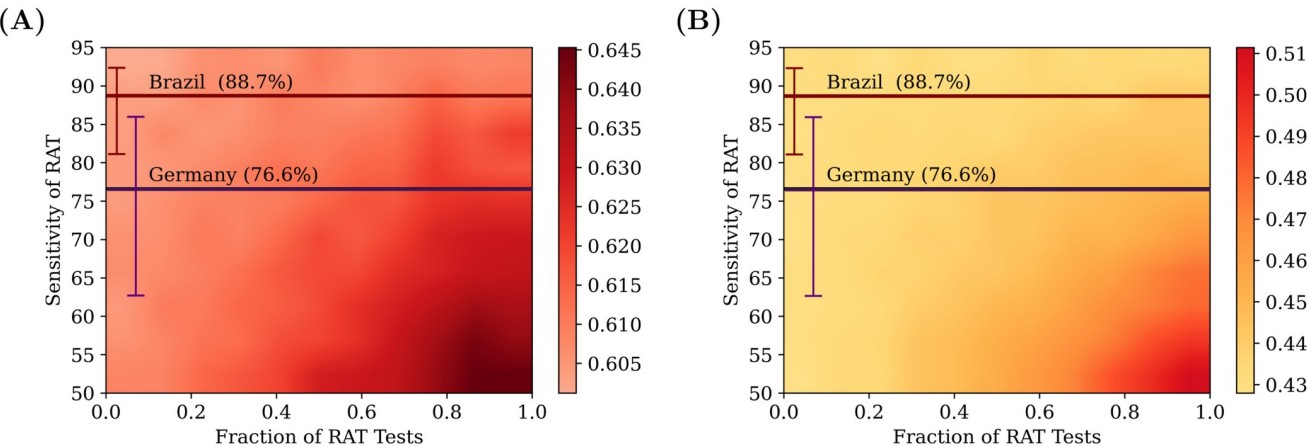

**Fig 6. Comparing different test mixtures.** Heatmaps comparing the effects of different test mixtures for two daily testing rates: 0.1% (left) and 0.5% (right), with testing starting when the recovered fraction reaches 20% of the population. The colours represent the total fraction of the population that had contracted the disease at some time. The *x*–axis represents the fraction of RAT tests in the testing mixture used, and the *y*–axis represents the test sensitivity of the RAT in the mixture. The horizontal lines represent the sensitivities of the SD Biosensor STANDARD Q COVID-19 Ag Test from surveys conducted in Germany and Brazil [46]. The specificity of the RAT is kept at 98%. At lower RAT sensitivities, there is a benefit of using a larger fraction of PCR tests, however as the RAT sensitivity is increased, this benefit is soon lost. The tests were assumed to have no delay between sampling and declaring of results; individuals who tested positive were confined to their homes and assumed to be ten-fold less infectious. However, homes were not quarantined, and other family members were allowed to move to work. 100 simulation runs of 200 days each were averaged over.

testing rate. However, small but significant improvements can also be made by changing the quarantining strategy to not just isolate the individual who tested positive, but also to quarantine their home and family members for 14 days (see Fig 8A).

The benefit of greater daily testing is, however, offset when we include a test delay for the PCR tests between the test sample being taken and the result being declared. Fig 8B shows that a delay of 2 days is enough to ensure that results for the suppression of infections using only PCR tests are no better than for an only RAT regime, while a larger delay of 5 days in Fig 8C ensures that analogous results for only PCR tests are significantly worse than for RATs. While larger daily testing rates still lead to an overall reduction in the total fraction of infected, the effect is much smaller than in the case of no delay. In this case too, quarantining the entire house generates an improvement, however the relative improvement is quite significant, given the low benefits of ramping up testing.

In the case of larger test delays, we considered a further quarantine protocol, in which we isolate the individual and quarantine the household from at the moment the test sample is taken, without waiting for the result to be announced. If the result is negative, both the individual and their homes are released. However, if the result is positive, both the individual and their homes are isolated and quarantined for a further 14 days. In this case, we make the following two interesting observations (see Fig 9):

(a). Quarantining houses when the sample is taken is a better option than simply ramping up testing, leading to lower total fraction infected.

(b). The benefit of using different test mixtures is nearly lost with delays: all test mixtures give rise to roughly the same total infected fraction.

The downside of this strategy of course is the larger number of individuals and homes that need to be isolated and quarantined (see S5 Appendix), which can result in severe consequences for certain sections of the population, as was evident during India's stringent lockdown.

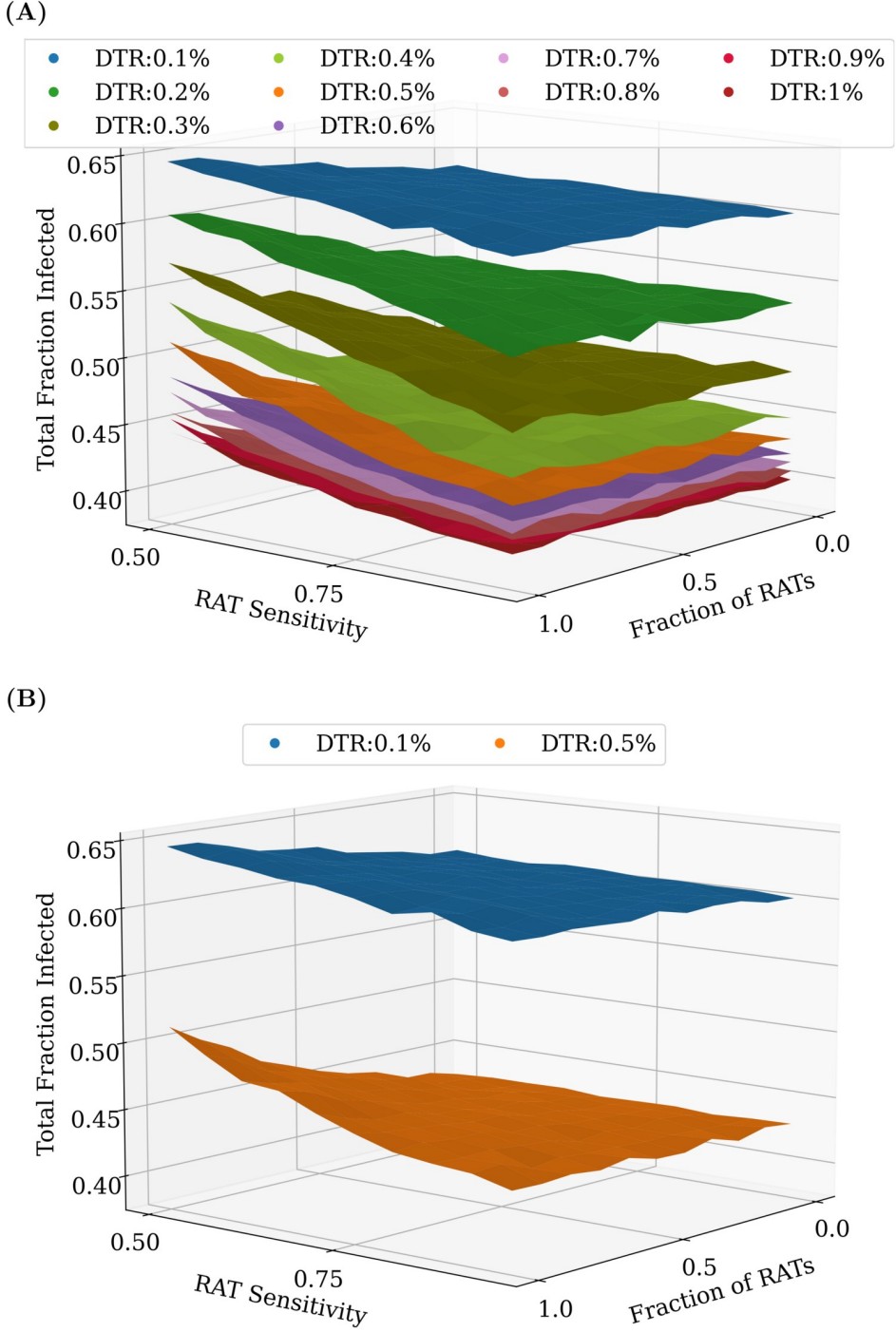

**Fig 7. Comparing different test mixtures across a range of testing rates.** An increase in the overall daily testing rate causes a large decrease in the total fraction of infected. (A) shows the effect of varying testing rates between 0.1% and 1% of the population per day, while (B) only shows 0.1% and 0.5% daily testing rates (the same as in Fig 6). In most cases, even at RAT:PCR mixtures of 80:20, good results can be obtained. As before, the plots were obtained by averaging over 100 simulation runs of 200 days each.

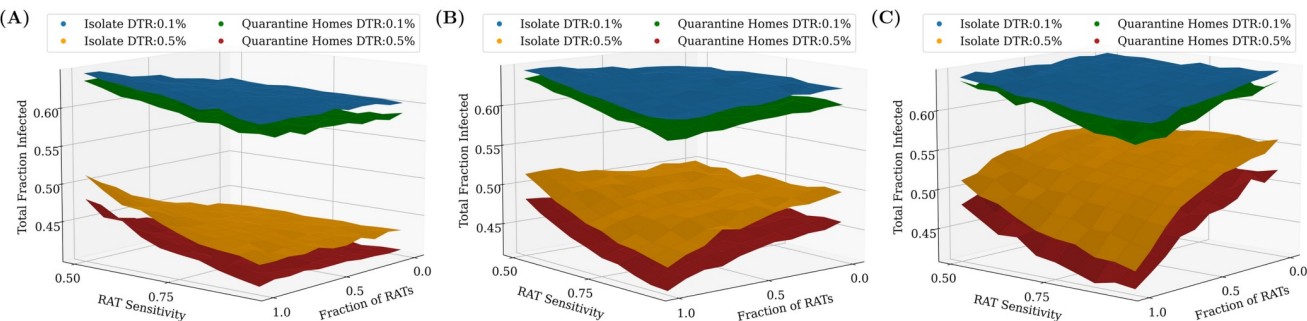

**Fig 8. Effects of PCR test delays and quarantining homes.** Testing is started when 20% of the population has recovered. (A) No PCR delay. Quarantining the home locations of the individuals in addition to isolating them at home is seen to have a small but not-insignificant benefit when there is no test delay. (B) 2 day PCR delay. When the PCR delay is increased to 2 days, the relative benefit of having PCR tests in the mixture is seen to disappear. (C) 5 day PCR delay. As the delay is further increased, the trend becomes clear: using even weak point-of-care RAT tests would be preferable to PCR tests with high delays. Additionally, when there are PCR delays, the benefit of increased testing is greatly diminished. In this case, however, the improvement that comes from quarantining of homes is much more significant at higher testing rates. As before, the plots were obtained by averaging over 100 simulation runs of 200 days each.

Fig 10 shows the effect of these different testing and quarantining strategies on both the total fraction infected as well as the *peak* infected fraction. While the former is a measure of the overall spread of the infection, the latter is important to understand whether health care systems have the capacity to handle the infections through the course of pandemic spread.

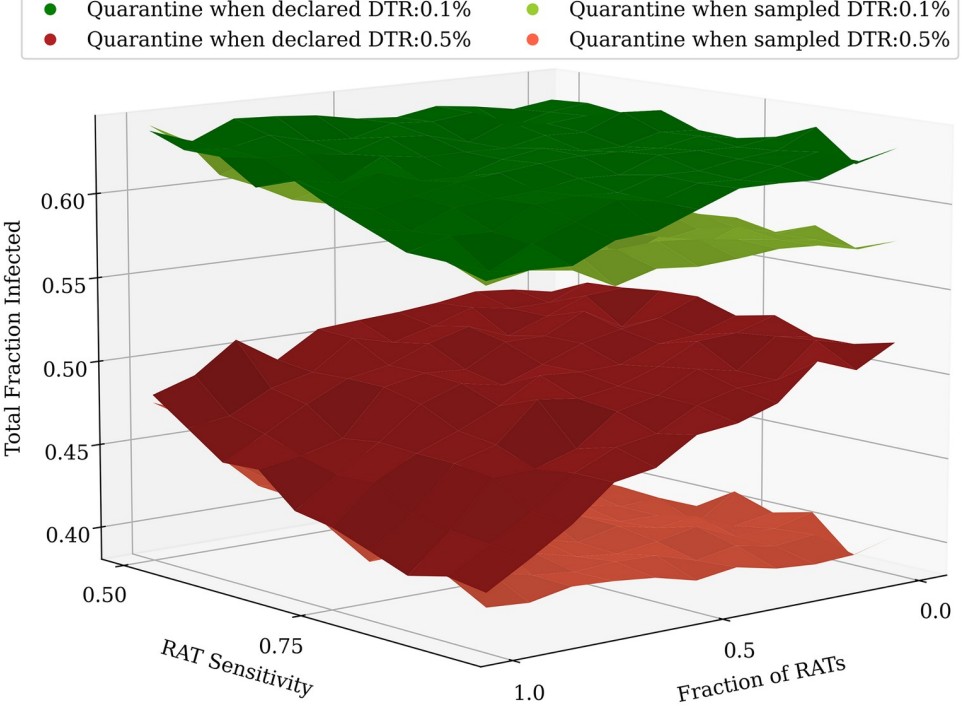

**Fig 9. Effects of quarantining when the sample is taken.** Including a larger test delay of 5 days for PCR tests while keeping RAT as point-of-care tests diminishes the benefit of having more PCR tests in the mixture, even when their homes are quarantined, as shown in Fig 8C. However, a new quarantining strategy where all tested individuals and homes are quarantined upon taking the test sample until the result is declared can be used to make up for this. The results were found to be qualitatively similar for a delay of 2 days as well, although the relative benefit of quarantining individuals at home was slightly offset by the lower test delay. The plots were obtained after averaging over 100 simulation runs of 200 days each.

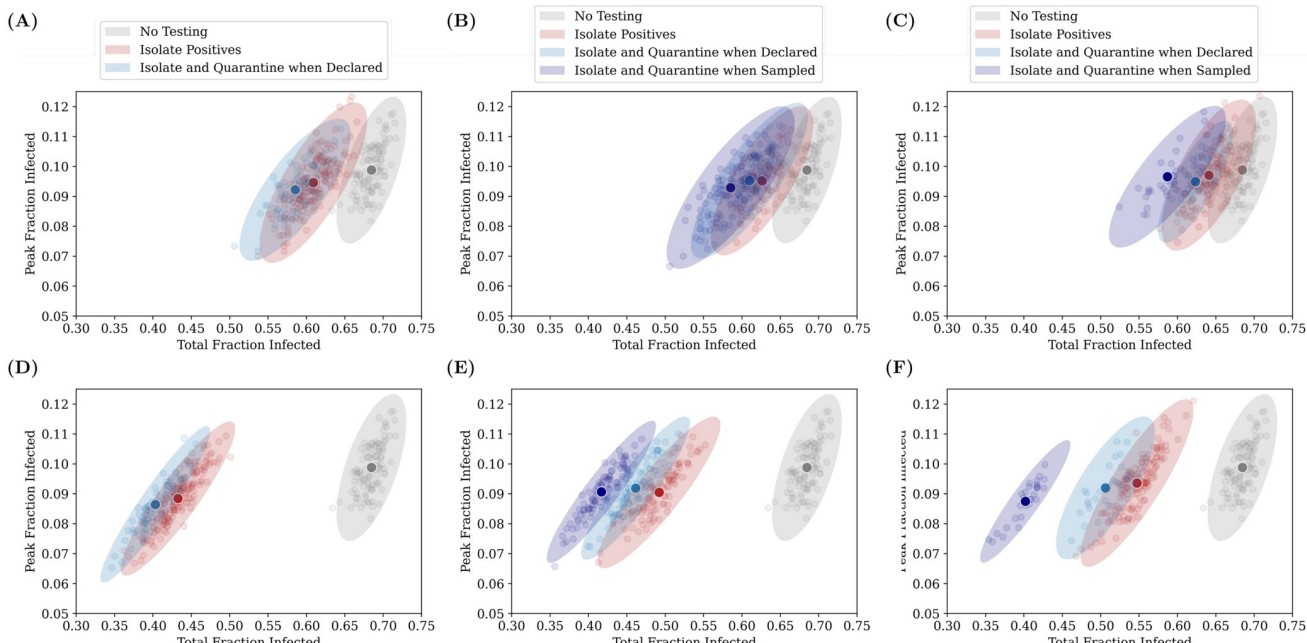

**Fig 10. Relative benefits of different quarantining strategies.** Each point on the graphs represents the result of a single simulation run for an RAT:PCR ratio of 80:20, and an RAT sensitivity of 75%. The darker coloured points represent the centre of mass of the points, and the ellipse radii represent three standard deviations from the mean in each direction. Ideally, we would require the peak and the total infected fraction to both be as low as possible, and therefore results closer to the bottom-left are "better" than those closer to the top-right. The top panel represents the case for a daily testing rate of 0.1%, while the bottom panel represents 0.5% testing daily. The panels on the left represent tests which have no delay (the result is declared immediately), while those in the middle and on the right represent mixtures in which PCR tests have a 2 day and 5 day delay respectively (RAT results are obtained instantaneously). The three quarantining strategies (see Methods) are compared—we find that quarantining when the test sample is taken is the most effective strategy. 100 simulation runs were conducted for each strategy, for a duration of 200 days.

Each point represents the result of a single simulation. The ellipses are centered around the average (highlighted darker point) and their radii represent 3 standard deviations of the distribution across simulations.

We find that increasing daily testing rate and/or isolating individuals who have tested positive or quarantining their homes has more of an effect on the total infected fraction than on the peak infected fraction, irrespective of whether there is a delay in declaring the results of tests. This is because we start testing when 20% of the population has recovered. If we start testing earlier, then we find significant reductions in the peak infected fraction also (see Fig 5 and S6 Appendix).

## Cost benefit analysis of different test mixtures

We now assume that we would like to achieve a some target fraction of recovered individuals (equivalent to the total number of infected individuals) at the end of the pandemic, and then ask which strategy might provide the most cost-effective method to achieve this. In Fig 11, the dependence of the daily testing rate required to attain a total infected fraction of 50%, on the RAT sensitivity and specificity is shown: pure RAT mixtures containing tests with low sensitivity (∼50%) require nearly twice the daily testing rate of pure PCR mixtures to attain the same target.

Once the daily testing rate needed to attain a certain target is known, the cost of attaining that target, given a test mixture with an RAT of a particular sensitivity, can be calculated, as

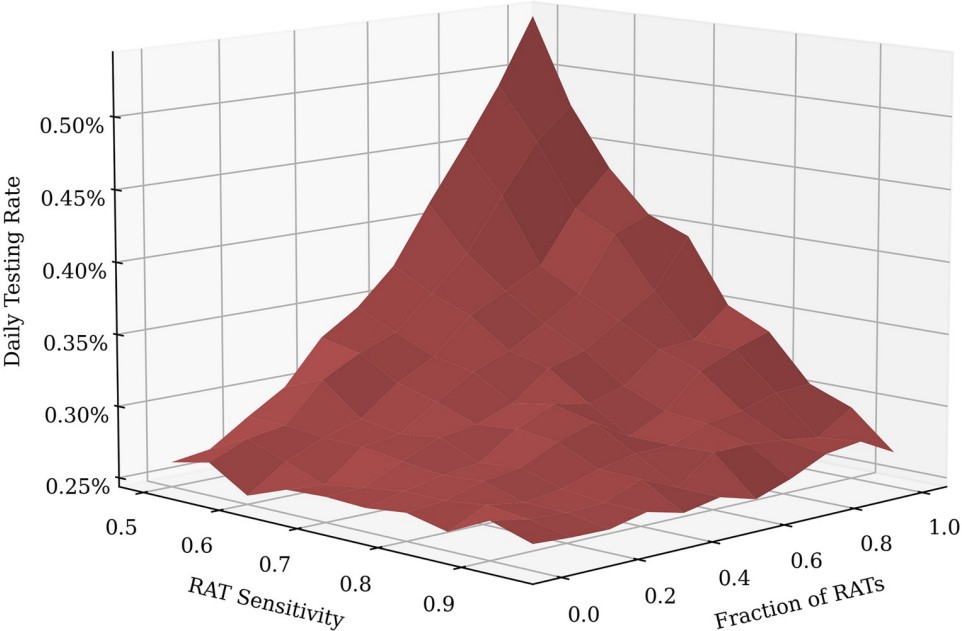

**Fig 11. Daily Testing Rates required to attain a target total fraction infected of 50% for varying RAT sensitivity and fraction.** As expected, when the test sensitivity is low and the test mixture is pure RAT, the required daily testing rate is largest, since much more testing is needed to compensate for the poor sensitivity. As before, the data for the above plot was obtained from 100 simulation runs of 200 days each.

shown in Fig 12. For a given fraction *f* of RAT tests in a mixture, the cost is calculated by

$$\text{Daily cost} = (\text{RAT}_{\text{cost}}f + \text{PCR}_{\text{cost}}(1-f))N_{\text{tests}}.$$

In all cases, we find that the optimum, cost-wise, is always either to use only RT-PCR tests or to use only RAT tests. PCR tests are only favoured when their relative cost is small enough and the RAT sensitivity is also low (see Fig 12A). If the relative cost of PCR tests exceeds a threshold value, then irrespective of the RAT sensitivity (within the range we considered) it is optimal to use only RAT tests to achieve the desired target total infected fraction.

If there are additional constraints, such as other external factors that limit the possible daily testing rate or unavailability of high sensitivity RAT tests, then the location of the optimum strategy might change, but given the shapes of the landscapes in Figs 11 and 12 the optima should still either consist of only PCR or only RAT tests.

## Synergy between non-pharmaceutical interventions and testing

We have previously shown (Fig 4) that increasing testing rates reduces the total fraction infected. However, there exists a threshold beyond which returns from increased testing begin to diminish. This threshold depends on how early in the spread of the disease we begin testing (see Fig 13A). We can thus ask whether these thresholds would change if, in addition to testing and isolation of positives, other non-pharmaceutical interventions (NPIs), such as masking, were introduced to reduce transmission.

We therefore explore the effect of combining testing with NPIs, the latter being modelled as an effective decrease in the infectivity parameter $\lambda_S$. In particular, we asked whether the combination of testing and NPIs is better than either separately, and whether NPIs can make up for starting later in the infection.

(A)

(B)

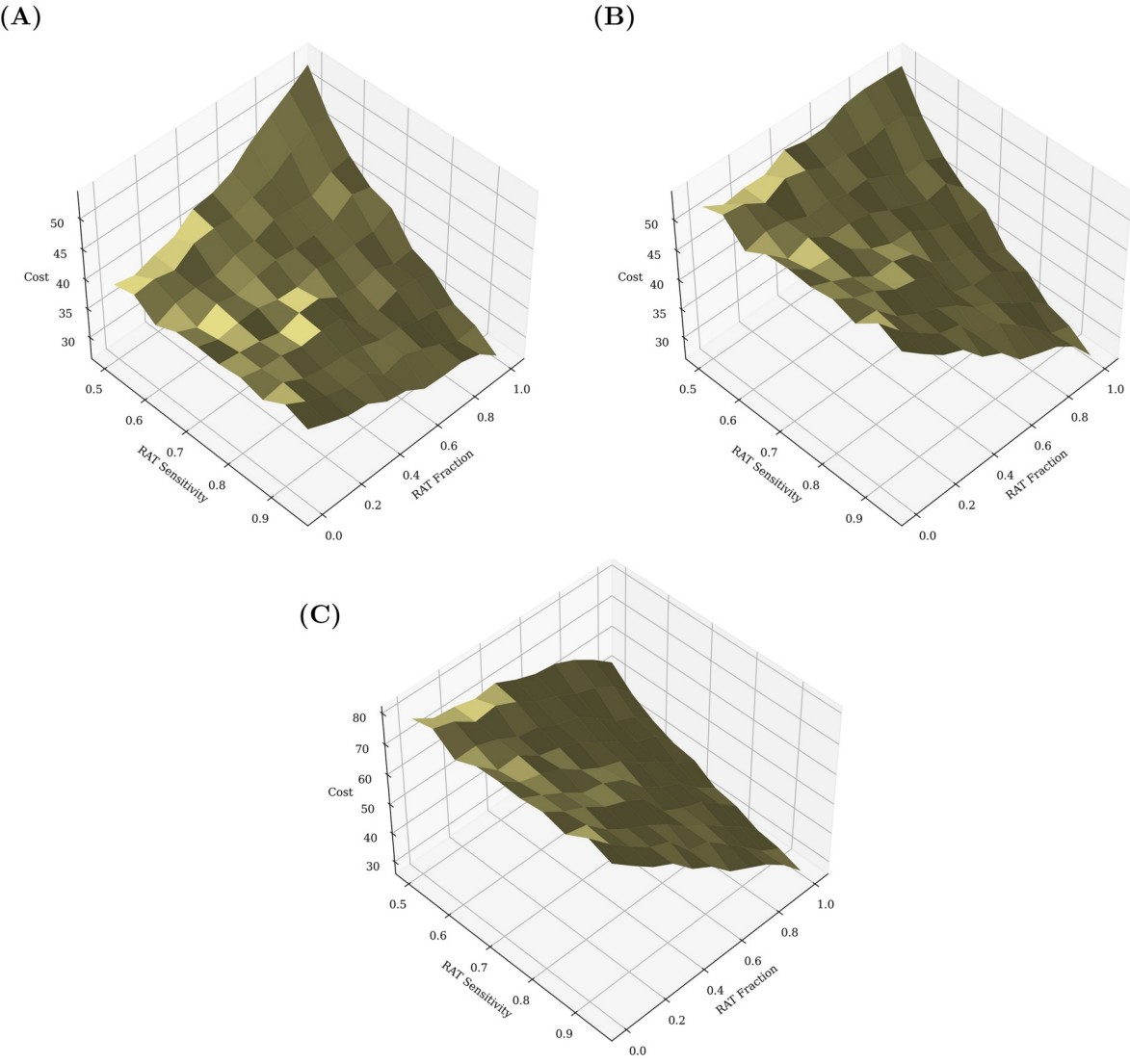

(C)

**Fig 12. Cost benefit analysis of test mixtures.** Comparisons of the total daily cost, taking into account test sensitivity and fraction of RAT and PCR tests in mixture, as well as the daily testing rate in order to attain a target recovered fraction of 50% of the population. The daily cost is measured in units of the cost of a single RAT. (A) Cost (PCR/RAT) = 1.5. (B) Cost (PCR/RAT) = 2. (C) Cost (PCR/RAT) = 3. Clearly, in all of the above cases, the highest cost occurs when the RAT sensitivity is lowest (0.5). At this point, depending on the relative cost of PCR to RAT, it may be more advantageous to choose a pure PCR or a pure RAT mixture. The optimal mixture may then be decided using other parameters, for example the largest attainable daily testing rate. At higher RAT sensitivities, typically it would be cheaper to attain the same target fraction of infected using a pure RAT mixture, despite the higher daily testing rate that would require.

We find that NPIs reduce the overall number of individuals who contract the disease, and that the threshold daily testing rate at which diminishing returns set in is lowered (see Fig 13B). Fig 13B also shows that there is indeed a synergy between NPIs and testing—at testing rates below the threshold, the total infected fraction produced by testing and NPIs is lower than the expected fraction than if the effects were simply additive (see S7 Appendix).

## Discussion

In this paper, we have presented a model for COVID-19 testing in a low-resource situation, where economic considerations as well as intrinsic limitations of different categories of tests

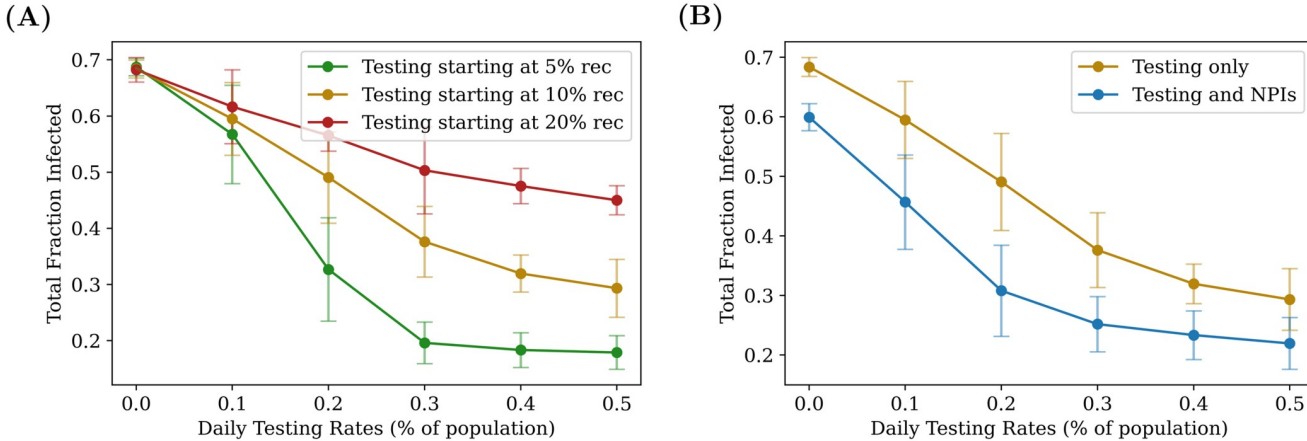

**Fig 13. Synergy between NPIs and testing.** Comparing the effects of two different types of interventions, given an RAT sensitivity of 75%, and an RAT:PCR ratio of 80:20. On the left, the total number of people recovered is plotted as function of daily testing rate: for a later start, testing has a weaker effect, and there exists a "threshold" (here roughly 0.3%, for the green curve) beyond which diminishing returns are gained from increasing testing. On the right, testing with and without NPIs are compared, when the interventions are both turned on when 10% of the population have recovered. NPIs are assumed to bring about a global reduction of 8% in the transmission parameter $\lambda_S$. The fact that the two curves initially separate as daily testing rate is increased, shows that the two interventions act in synergy (see also S7 Appendix). As before, the plots were obtained by averaging over 100 simulation runs of 200 days each.

dictate that a mix of different types of tests be used rather than a single type. We considered the case of a combination of a relatively inexpensive but less sensitive point-of-care rapid antigen test (RAT) with a more sensitive but also more expensive RT-PCR test. We showed that the use of just RAT tests could yield epidemiological outcomes comparable to those obtained through RT-PCR-based testing, in terms of reducing both the peak numbers of infected and the total infected by the end of the epidemic. We assessed optimal testing regimes, taking into account test sensitivity and specificity, background levels of seroprevalence and current test pricing in the states of India. We found that even 100% RAT test regimes should be acceptable, from both an epidemiological as well as a economic standpoint, provided a number of conditions were met.

Intuition for our results can be obtained by observing that the effectiveness of any testing strategy depends on whether the number of tests administered per day are sufficient to locate all the *new* cases each day. A testing rate of 0.5% will be effective in suppressing the epidemic if the number of daily new cases is less than 0.5% of the population (for purely PCR tests) or ≈0.6% (for pure RAT with 80% sensitivity), which explains why we see reasonable results with RAT:PCR mixtures. This intuition provides an easily estimated upper bound on the required testing rate; in fact the tests need only catch enough of the new cases to bring the reproduction number below 1, but that is harder to estimate. Variation in the infection model parameters which result in more asymptomatic cases would reduce the number of cases caught and increase the required testing rate. Similarly, starting testing at 20% recovered has less of an effect than starting at 10% recovered because both situations occur on the upswing of the infection spread and the former has more than twice the number of new cases appearing per day. This also explains the synergy between non-pharmaceutical interventions, which reduce the number of new cases per day, and the testing strategies we examined.

How closely do our assumptions resemble reality? Current ICMR-recommended testing protocols in India list a number of different categories for which testing is required [47]. First, for routine surveillance in containment zones and screening at points of entry, where all symptomatic cases, including health care workers and frontline workers, are required to be tested. In addition, all asymptomatic direct and high-risk contacts of a laboratory-confirmed case

within a few days of contact, together with all asymptomatic high-risk individuals in containment zones, are to be prioritized for testing. The order of priority is, in sequence, an RAT and an RT-PCR test (or TrueNat or CBNAAT in place of the RT-PCR test). Second, routine surveillance in non-containment areas involves the testing of all symptomatic cases with a history of international travel in the last 14 days, testing of symptomatic contacts of a laboratory-confirmed case as well as of symptomatic health care workers and frontline workers who are involved in containment and mitigation activities. The RAT is recommended here as the first choice of test. It is only in hospital settings, for patients with SARI (Severe Acute Respiratory Infection), symptomatic patients presenting in a healthcare setting, asymptomatic high-risk patients who are hospitalized or seeking immediate hospitalization such as immuno-compromised individuals and a number of related categories, that individuals are to be tested first by RT-PCR and then by RAT.

These protocols prioritize the testing of symptomatic individuals, as we have assumed here, and largely, also as assumed here, use the RAT test as a first test of choice [47]. These are consistent with the modeling choices we make in this paper. We also note that our focus is on the epidemiological consequences of testing strategies. Thus, in our model we do not explicitly consider testing of individuals who are already hospitalized, and the testing rates we report should be interpreted accordingly.

For simplicity, we have used a perfect RT-PCR test as a point of comparison in our simulations. In reality, the sensitivity of RT-PCR tests in the field have been reported to be in the range 85—95% [48]. This will only reinforce our conclusion that using a high percentage of RAT tests can nevertheless provide good epidemiological outcomes. Pekosz et al. [49] find that for patients within 7 days of onset of symptoms, RAT results correlate with the presence of culturable virus better than RT-PCR—this would also strengthen our conclusion. It has also been reported that doing two consecutive RAT tests on the same patient raises the sensitivity to 99% [50]. The effect of such a strategy in our models can be easily inferred from Figs 7–10; our results suggest that the gain from a higher sensitivity does not overcome the effective halving of the testing rate.

We used a multiplier to quantify the relative costs of the RAT and the RT-PCR tests. The costs discussed below are explicitly for tests at private facilities. (Testing is offered free of charge at government testing facilities, so the associated costs fall on the state. We will assume that these costs parallel those outside the government system.) Current costs of RT-PCR testing in the state of UP are between INR 700 ($9.5) and INR 900 ($12.28), down from INR 4,500 ($61.38) in private laboratories at the beginning of the pandemic [51]. The state of Bihar has capped its RT-PCR rates at INR 800 ($10.91), while RAT tests cost INR 400 ($5.46) [52]. The state of Orissa, as of Dec 3, offers RATs at INR 100 ($1.36 while RT-PCR test costs are capped at INR 400 ($5.46) [53]. (The Supreme Court of India is currently hearing a plea that asks to have the cost of RT-PCR tests capped at INR 300 ($4.03) across India.).

These rates are somewhat above the threshold where an all-PCR regime is equivalent to or only slightly more expensive than a mixed regime. As we point out here, the most important determinant of controlling a pandemic at intermediate levels of seroprevalance is simply the amount of testing that is possible. It is here that we expect that all-RAT regimes may make more economic sense, while returning results equivalent to all-RT-PCR-test regimes. We have pointed out the importance of reducing delays between tests and their results being available. Such delays decimate any advantage that RT-PCR tests enjoy *vis a vis* RATs.

Our results focus primarily on the role of testing at intermediate levels of seroprevalence, around 20%. For our parameters, in the absence of testing, this roughly coincides with the peak of the infection. For completeness, in S6 Appendix we show results when testing started at 5% and 10%. Qualitatively, the results are similar. As expected, the earlier one starts testing,

the lower the testing rate needed, but the relative impact of different strategies is similar, although quarantining of homes has a more pronounced effect when starting at 5% seroprevalence. We have also assumed a step change in daily testing rate, from zero to a constant value when the population reaches a threshold seroprevalence. In reality, testing would ramp up at a certain rate. From the heatmap stacks in Fig 7 one can infer that while the number of tests per day are lower than the daily number of new cases, testing rate has a close-to-linear effect on the epidemic spread, saturating once the tests per day become close to the daily number of new cases. However, the RAT:PCR ratio has less of an impact. Therefore, our conclusion that lower sensitivity of tests is less important than the coverage would hold even if testing ramped up slowly.

Our results are consistent with recent work which found that the frequency of testing was the main determinant of epidemic control at the level of populations, with only marginal improvement being afforded by the use of a more sensitive test [54, 55]. Our aim in this paper was more specific, though, in that it examined a scenario for testing that is the one used across India and has been the topic of much discussion. Our focus is on mitigation rather than elimination strategies, aiming at reducing the total numbers of infected to the vicinity of a pre-decided target while quantifying the trade-offs required to do so. This may be a more realistic approach in resource-constrained situations at intermediate stages of the pandemic.

We point out several caveats: Network models provide an individual-based representation of disease dynamics, adding a layer of stochasticity to purely deterministic epidemiological models such as the SEIR model. However, this stochasticity enters in multiple ways, ranging from the choice of network to the probability distributions governing the times spent in each state (compartment). We made relatively simple choices for these, in the absence of more specific information about interaction networks relevant to COVID-19 spread in a evolving pandemic within a low-income setting. We also experimented with different networks, with varied degrees of super-spreading, to ensure that our results remained qualitatively robust. For simplicity, we chose simple exponential distributions of residence times in each compartment; these could be generalized to the biologically more relevant log-normal distributions in a straightforward manner.

Our calculation of the costs of different testing regimes were based simply on the relative costs of administering an RAT vs a PCR test. We assumed that this differential remained fixed, irrespective of the quantum of testing and did not change with time. However, technological improvements, competition and state intervention have helped to narrow the price gap between RAT and RT-PCR tests considerably in the past few months. It remains to be seen whether this gap will shrink further.

We also do not address the issue of combining testing and quarantining strategies with contact tracing. While we do target tests towards symptomatic individuals, the question arises whether contact tracing could usefully identify asymptomatic individuals and thereby dampen the pandemic more effectively. Our strategy (b) of quarantining entire homes when an individual tests positive may be considered a very crude form of contact tracing, limited only to the individual's family members. In fact, this works only marginally better than strategy (a) which only isolates the individual (see Fig 10). Ref. [56] explores the effect of quarantining combined with more sophisticated contact tracing algorithms on epidemics spreading on networks, but does not consider RAT tests or the effect of starting at intermediate seroprevalences. It would be interesting to combine these approaches in future models.

Finally, our calculations address the contentious issue of whether a testing policy that is based on having a larger fraction of rapid, less sensitive tests can at all be justified. We show that it can, demonstrating this result within a individual-based network model that can be expected to provide qualitative insight. Our work supports the idea that population-level

coverage is more important than test sensitivity, also pointing to a non-linear synergy between non-pharmaceutical interventions such as mandatory mask wearing, with testing accompanied by isolation and quarantine [57].

Although vaccine rollout began in India on January 16, 2021, only about 130 million have so far been vaccinated and, of that, only about 17 million have received two doses. As of the time of writing (April 23, 2021), testing in India has faltered, with only about 1.6 million tests being conducted each day. Test positivities for India as a whole currently exceed 17%. It is in this context that such questions of optimal testing remain central to India's response to the pandemic. They are also central to similar questions in LMIC settings.

## Supporting information

**S1 Appendix. Analysis of Compartmental Model.** The dynamics of the disease-progression in a single well-mixed compartmental model are studied, and the reproductive ratio is calculated by applying a next-generation matrix method to the governing equations. The result is that for a single well-mixed compartment, the reproductive ratio $R_0$ is

$$R_0 = \lambda_S \left( \frac{\gamma}{\lambda_A} + (1 - \gamma) \left( \frac{1}{\lambda_P} + \frac{\delta}{\lambda_{MI}} + (1 - \delta) \left( \frac{1}{\lambda_{SI}} + \frac{\sigma}{\lambda_H} \right) \right) \right).$$

Using the parameters given in Table 1, this yields $R_0 = 2.374$.
(PDF)

**S2 Appendix. Scaling up our simulation.** The results of the main text are obtained for a population size of 10,000 individuals, with 2,750 locations (2,500 home locations and 250 work locations). This corresponds to a Poisson distributed population in the home and work locations with a mean of 4 and 40 individuals respectively. We also started with an initial infection seed of 10 asymptomatic individuals (0.1% of the population). However, we also tried a number of runs that were scaled-up to larger population sizes (while keeping the initial infection fraction the same, and preserving household and work location sizes) in order to check for finite-size effects. The results of these simulations are shown in the figure, and they are found to agree remarkably well with our results for 10,000 individuals, indicating that this problem scales well with the total population.
(PDF)

**S3 Appendix. Effects of random testing.** For random testing, available tests are distributed among eligible individuals in the population completely randomly, with no preference given to symptomatic individuals. The figure shows the effects of purely random testing, demonstrating the importance of first targeting symptomatics to reduce the total number of infections over the course of the pandemic.
(PDF)

**S4 Appendix. Effects of test delays on quarantining strategies.** As shown in the main text, the benefit of the PCR tests' sensitivity is offset by the introduction of the delay. However, this can be countered by quarantining individuals or homes when the samples are taken. When large fractions of RAT tests are used (RAT:PCR $\sim$ 80:20) equivalent or marginally better results can be obtained by quarantining the homes of the tested individual when they are declared positive, instead of isolating them when they are sampled.
(PDF)

**S5 Appendix. Cost of imposing interventions when test sample is taken, rather than when test results are declared.** Strategies that involve interventions being enforced when the

individual is sampled are found to require a larger number of people and homes confined. We consider the case when only PCR tests are used with a delay of 5 days between sampling and the results being declared, since this leads to the largest number of people and homes quarantined. In the case of interventions that are enforced when the individual is sampled for a test, the number of people or homes confined does not die out even after the pandemic has passed, since people continue to be tested and remain confined for a duration of 5 days until the PCR result is declared. The peak number of people confined varies from 4% to 6%, and this translates into a fraction of homes quarantined between 15% and 25%.
(PDF)

**S6 Appendix. Effects of starting testing earlier.** As discussed in the main text, the majority of our simulations were carried out for the case where testing is begun when 20% of the population has recovered from the disease. As expected, the earlier one starts testing, the lower the testing rate needed for the same reduction in total infections. However, the relative impact of the different testing and quarantining strategies is similar, although quarantining of homes has a relatively more pronounced effect when starting at 5% seroprevalence.
(PDF)

**S7 Appendix. Synergy between testing and masking is more efficient than purely additive or multiplicative effects.** As shown in the main text, there exists a synergy between testing and masking, in excess of a simply additive or even multiplicative effect (see below). In the figure, we compare the effect that masking would have had if the effect had been purely additive or multiplicative. For a given daily testing rate of $r$, let $f(r)$ describe the effect of testing (without masking), and $g(r)$ describe the effect of testing and masking combined. We define $\Delta N = f(0) - g(0)$ to be the effect of purely masking the population. Then, by an additive effect we mean that:

$$h_{\mathrm{add}}(r) = f(r) - \Delta N,$$

and by a multiplicative effect we mean that

$$h_{\mathrm{mult}}(r) = f(r) \times \left( \frac{g(0)}{f(0)} \right).$$

All four functions $f(r)$, $g(r)$, $h_{\mathrm{add}}(r)$, and $h_{\mathrm{mult}}(r)$ are shown in the figure.
(PDF)

**S8 Appendix. Flowcharts describing the schematic of the numerical calculations.**
(PDF)

## Acknowledgments

We are grateful for stimulating discussions with Harish Iyer and Kayla Laserson.

## Author Contributions

**Conceptualization:** Philip Cherian, Sandeep Krishna, Gautam I. Menon.

**Data curation:** Sandeep Krishna, Gautam I. Menon.

**Formal analysis:** Sandeep Krishna, Gautam I. Menon.

**Funding acquisition:** Sandeep Krishna, Gautam I. Menon.

**Investigation:** Philip Cherian, Sandeep Krishna, Gautam I. Menon.

**Methodology:** Philip Cherian, Sandeep Krishna, Gautam I. Menon.

**Project administration:** Sandeep Krishna, Gautam I. Menon.

**Resources:** Sandeep Krishna, Gautam I. Menon.

**Software:** Philip Cherian, Sandeep Krishna.

**Supervision:** Sandeep Krishna, Gautam I. Menon.

**Validation:** Philip Cherian, Sandeep Krishna, Gautam I. Menon.

**Visualization:** Philip Cherian, Sandeep Krishna, Gautam I. Menon.

**Writing – original draft:** Philip Cherian, Sandeep Krishna, Gautam I. Menon.

**Writing – review & editing:** Philip Cherian, Sandeep Krishna, Gautam I. Menon.

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
