## [Decision Letter · Decision Letter 0]

1 Apr 2021

Dear Prof. Menon,

Thank you very much for submitting your manuscript "Optimizing testing for COVID-19 in India" for consideration at PLOS Computational Biology.

As with all papers reviewed by the journal, your manuscript was reviewed by members of the editorial board and by several independent reviewers. In light of the reviews (below this email), we would like to invite the resubmission of a significantly-revised version that takes into account the reviewers' comments.

I agree with reviewer 1 that the final paragraph of the introduction would be better suited in the discussion.

We cannot make any decision about publication until we have seen the revised manuscript and your response to the reviewers' comments. Your revised manuscript is also likely to be sent to reviewers for further evaluation.

Sincerely,

Benjamin Muir Althouse

Associate Editor

PLOS Computational Biology

Virginia Pitzer

Deputy Editor-in-Chief

PLOS Computational Biology

I agree with reviewer 1 that the final paragraph of the introduction would be better suited in the discussion.

Reviewer's Responses to Questions

**Comments to the Authors:**

Reviewer #1: The authors have developed a network-based model to assess different diagnostic approaches to COVID-19 in India. This is an important question to explore through mathematical modeling and relevant to all geographies – India and elsewhere. Diagnostics are one of the three primary interventions that can help reduce the morbidity and mortality associated with COVID-19. The manuscript is very clearly and concisely written. The methods are very clearly described and seem perfectly appropriate to address the question raised by the authors. I have one overarching comment and several minor comments. Otherwise, this is an important manuscript and will contribute to ongoing efforts to address the epidemic in India and elsewhere.

MAJOR COMMENT

• The authors have assumed 100% sensitivity for RT-PCR and varying levels of sensitivity for rapid antigen tests. The discussion/conclusions could be strengthened by acknowledging recent evidence that brings this assumption into question. In a study published by Pekosz, et al. in Clinical Infectious Diseases (CID) indicates that rapid antigen testing correlates with culturable virus, indicating that viral particles can be transmitted when someone tests positive by such tests; whereas RT-PCR does not correlate with culturable virus. The positive predictive value of antigen tests was considerably higher than RT-PCR. This has been attributable to the sensitivity for viral RNA rather than detection of surface protein. This could give rapid antigen tests and even greater advantage over RT-PCR. I think it would be difficult for the authors to incorporate this point in their existing model; however, a recognition of this point and the directionality of the potential bias introduced into their model mentioned somewhere in the manuscript could be helpful for the reader.

• In addition, I am aware that in many settings where rapid antigen tests have been used, a testing algorithm has been employed. So, for example, for a SARI or ILI patient, a rapid antigen test is first used. If the patient is positive, then they are considered positive. However, if a test is negative, given they are then tested by RT-PCR. Is this happening at all in India? Another interesting paper found that a commonly used rapid antigen test found that using two tests consecutively increased the sensitivity to ~99%. Slovakia, admittedly with only 5.4 million people, tried to test the entire country at once. Might the authors be able to look into comparing such testing algorithms? Or at least mentioning them as additional points to explore in future research?

MINOR COMMENTS

Introduction

• The authors cite a Wikipedia article when stating that that the stringently enforced lockdown in India contributed to slow epidemic growth. Surely there must be more appropriate references for this.

• The final paragraph of the introduction beginning on line 93 provides an interpretation of the findings. I will defer to the editors regarding whether this is status quo for this journal. It would seem more appropriate as part of the discussion section. However, if the editors indicate that it should remain in the introduction, it would be important to include some of the limitations of the model findings in this paragraph.

Model and methods

• In table 1, it would be helpful for readers if the authors could provide the epidemiologic definitions for each of the model parameters.

• Beginning on like 167, the authors note that health care workers are 25-30% more likely to become infected compared with non-health care workers in the absence of testing. Is this based on the observations made from the model? Or from empiric evidence? If the latter, more details would be helpful. If from the former, I would suggest including in the results rather than the methods.

Models for testing and quarantining

• The authors start testing after 20% of the population has been recovered. What is the justification for this? Testing in India has scaled up relatively fast and test positivity in most states remained below the 5% target likely before the 20% seropositivity. Different starting times are then presented in Figure 4. In reality, testing would be scaled up over time rather than once a certain threshold is attained. Can the authors comment on this?

• For eligibility of testing, did the authors consider asymptomatic individuals identified through contact tracing? If not, what are the implications of not including these individuals in the model?

• Beginning on line 196, the authors note that unused tests are distributed randomly among eligible individuals in the population. Why was this approach adopted? Surely, some of the states in India have used tests in random testing. But I would imagine this would be done using RAT rather than RT PCR. Can the authors expand on this approach?

Reviewer #2: This paper aims to model the consequences of using antigen vs PCR tests for COVID surveillance testing, specifically in the context of India. The paper assesses the consequences of varying the fraction of tests which are antigen vs PCR, as a function of the sensitivity of the antigen test. It is found that when the antigen test is at least moderately sensitive, incorporating a high fraction of antigen tests into the surveillance testing program is effective at reducing infections.

Assessing these specific tradeoffs in the design of surveillance testing programs is a timely and worthwhile goal, as is exploring them specifically in the context of India. Overall, the results seem fairly sensible. Some specific comments about individual parts of the paper:

(1) The model description is very unclear and hard to follow. First, a compartmental ODE model is presented. Then, an individual-level network simulation is introduced. What is the relationship between the ODE formulation and the individual-level, discrete-time network model? And, how is the force of infection (Eq. 9) formally used in the network model? If the network model is what is used throughout the paper, a complete formal description of it should be given somewhere in the paper.

(2) A sensitivity analysis would be helpful to test the impact of delays in reporting of PCR test results. It is unsurprising that 5 days is too long for the results to be useful, but what about an intermediate length of time?

(3) The model seems to assume that both antigen and PCR tests have uniform sensitivity over the course of an individual's infection? This is not consistent with available evidence. Of course some simplifying assumptions are reasonable, but some basis (e.g., sensitivity analysis) should be given for thinking that this does not impact the results too much.

(4) How many simulations do the results average over? It seems like some of the variation, particularly in the left-hand plot of Figure 6, might be due to random variation between runs.

(5) What is the rationale for including quarantine strategy C? It seems very difficult to motivate a policy which preemptively quarantines people who are randomly selected for surveillance testing, i.e, are asymptomatic and have no known exposures. I would recommend just removing this scenario.

(6) The description of Figure 13 as studying "masking" is a little strange since it is implemented as a generic reduction in the infectivity of the disease -- is this distinguishable in the model from the effect of any generic NPI, e.g. social distancing? I would describe this just as measuring the interaction between testing and any combination of other interventions which reduce beta.

(7) In the discussion, some of the results are intuitively motivated by asking what level of testing would allow all new cases to be located (line 372). I don't think that this is the right intuition. To drive cases to 0, we just need R < 1, not that every new case can be found. Also, the modeled policies are allocating many/most of their tests to random surveillance testing, which isn't targeted to exactly finding the newly infected people.

(8) How long is the simulation run for? Figure 6 just claims to report the fraction infected after "some time".

**Have all data underlying the figures and results presented in the manuscript been provided?**

Reviewer #1: Yes

Reviewer #2: None

PLOS authors have the option to publish the peer review history of their article (what does this mean?). If published, this will include your full peer review and any attached files.

Reviewer #1: No

Reviewer #2: No
---

## [Decision Letter · Decision Letter 1]

28 May 2021

Dear Prof. Menon,

We are pleased to inform you that your manuscript 'Optimizing testing for COVID-19 in India' has been provisionally accepted for publication in PLOS Computational Biology.

Best regards,

Benjamin Muir Althouse

Associate Editor

PLOS Computational Biology

Virginia Pitzer

Deputy Editor-in-Chief

PLOS Computational Biology

Reviewer's Responses to Questions

**Comments to the Authors:**

Reviewer #1: I thank the authors for their thoughtful and thorough responses to my questions and comments.

Reviewer #2: Thanks to the authors for addressing my comments. The manuscript is now much easier to follow. It makes a valuable contribution to the study of testing policy in India/LMIC settings.

**Have the authors made all data and (if applicable) computational code underlying the findings in their manuscript fully available?**

Reviewer #1: Yes

Reviewer #2: None

PLOS authors have the option to publish the peer review history of their article (what does this mean?). If published, this will include your full peer review and any attached files.

Reviewer #1: No

Reviewer #2: No

---

## [Editor Report · Acceptance letter]

28 Jun 2021

PCOMPBIOL-D-21-00099R1 

Optimizing testing for COVID-19 in India

Dear Dr Menon,

I am pleased to inform you that your manuscript has been formally accepted for publication in PLOS Computational Biology. Your manuscript is now with our production department and you will be notified of the publication date in due course.

With kind regards,

Andrea Szabo
